# Imbalanced Classification via a Tabular Translation GAN

## Abstract

When presented with a classification problem where the data exhibits severe class imbalance, most standard predictive methods may fail to accurately model the minority class. We present a novel framework based on Generative Adversarial Networks which introduces a direct translation loss in conjunction with optional cyclic and identity losses to map majority samples to corresponding synthetic minority samples. We demonstrate that this translation mechanism encourages the synthesized samples to be close to the class boundary. Furthermore, we explore a selection criterion to retain the most useful of the synthesized samples. We conduct extensive experiments on tabular class-imbalanced data, including huge datasets, using several downstream classifiers. These empirical results show that the proposed method improves average precision when compared to alternative re-weighting and oversampling techniques.

## 1 Introduction

Data that exhibits class imbalance appears frequently in real-world problems (Bauder et al., 2018), in varying domains and applications: detecting pathologies or diseases in medical records (Zhao et al., 2018), preventing network attacks in cybersecurity (Wheelus et al., 2018), detecting fraudulent financial transactions (Makki et al., 2019), distinguishing between earthquakes and explosions (Rabin et al., 2016) and detecting spam communications (Tang et al., 2006). In addition to these applications, where the class distribution is naturally skewed due to the frequency of events, some applications may exhibit class imbalance caused by extrinsic factors such as collection and storage limitations (He & Garcia, 2009).

Most standard classification models are designed around and implicitly assume a relatively balanced class distribution; when applied without proper adjustments they often converge to a solution that over-classifies the majority class due to its increased prior probability (Johnson & Khoshgoftaar, 2019). These models thus neglect recall on the minority class and lead to unsatisfactory results when we desire high performance on a more balanced testing criterion. This issue is exacerbated by the fact that commonly used metrics such as accuracy may be misleading in evaluating the performance of the model. For example, models that naively classify all samples as majority may have high accuracy under severe class imbalance. Most approaches to dealing with these shortcomings fall broadly into two categories: re-weighting the loss objective to more heavily account for the minority class, and resampling the input dataset such that the minority class is more prominent.

The proposed approach falls in the latter category, and provides a technique to oversample the minority class by generating synthetic minority samples that correspond to translations of real samples from the majority class. Leveraging the diversity of the real majority samples allows this translation approach to generate samples that are beneficial to classification performance.

We focus our attention on the problem of class-imbalanced binary classification for tabular datasets. We consider datasets to be tabular if their feature space is not inherently structured, in contrast to modalities where feature structure can be exploited by incorporating inductive bias in the model, such as visual and audio data. Many of the aforementioned real-world applications are instances of this problem. Additionally, these commonly occurring applications are all binary classification tasks; when a larger number of classes are involved, the desired application can sometimes be formulated as a binary classification task using the one-versus-rest method (Kang et al., 2017).

## 1.1 Contribution

In this work, our contribution is to achieve the following goals:

- Propose a novel GAN-based model (TTGAN) that is capable of generating synthetic minority samples for tabular data.

- Leverage the diversity of the majority class by ensuring that the generated points are translated from the majority class.

- Provide a strategy for selection of the most useful subset of these translated synthetic samples.

- Apply TTGAN on a variety of imbalanced tabular datasets with different characteristics.

- Demonstrate increased downstream balanced classification performance compared to alternative class-imbalanced techniques.

## 1.2 Related work

**Background**   Methods to address class imbalance have been extensively studied. A straightforward approach involves re-weighting the cost/penalty associated with the classes in proportion to their cardinality. Alternatively or in addition, resampling the input training data distribution is a common technique. The simplest forms of this technique are random oversampling, which duplicates samples from the minority, and random undersampling, which discards samples from the majority. Undersampling risks losing valuable information in the process, while oversampling introduces the risk of overfitting (Chawla et al., 2004).

**Oversampling**   Methods more sophisticated than random oversampling have been introduced in the last few decades, notably SMOTE (Chawla et al., 2002) and its variants, which generate new minority samples by interpolating nearby training samples. One such variant, Borderline SMOTE (Han et al., 2005) is similar to the proposed method in the sense that it oversamples specifically near the class boundary region. Kovács (2019) tests numerous SMOTE variants and identifies polynom-fit-SMOTE (Gazzah & Amara, 2008) as having strong overall performance. SUGAR (Lindenbaum et al., 2018) takes a geometric approach, generating new samples along a manifold learned through a diffusion process. Methods that aim to mitigate class imbalance specifically in the image domain by synthesizing samples have become popular in recent years. Mullick et al. (2019) and Choi et al. (2021) jointly train a classifier network with a GAN to generate synthetic samples near the class boundary. Lee et al. (2020) propose an additional GAN-based method for mitigating class imbalance on images, and further techniques are surveyed in (Zhang et al., 2021).

**Tabular synthetic data generation**   A unique challenge that may present itself when tasked with generating tabular data is that of modeling a mixture of continuous and discrete (categorical) features, with non-Gaussian distributions. TGAN (Xu & Veeramachaneni, 2018) uses an LSTM-based GAN to generate synthetic samples; CTGAN (Xu et al., 2019) uses a conditional generator. While our approach is not designed to tackle the problem of modeling discrete features, techniques inspired by these works may in principle be incorporated in our model as well. Darabi & Elor (2021) synthesize new data by interpolating points in the latent space of an autoencoder. Tabular data generation methods have been applied for multiple specific applications: medGAN (Choi et al., 2017) is a model designed to generate medical patient records and tableGAN (Park et al., 2018) introduce a model that synthesizes tabular data while preserving privacy. comfortGAN (Quintana et al., 2020) synthesizes samples in thermal comfort datasets to address class imbalance.

**Translation**   Unpaired, unsupervised translation techniques have become prevalent in the deep learning literature in recent years particularly for the visual modality (Zhu et al., 2017; Kim et al., 2017; Choi et al., 2018). Such techniques usually rely on a cyclic loss to maintain translation correspondence in absence of supervision. The model presented by Choi et al. (2018) is also able to perform the translation across more than two domains. These methods have been applied to translate images for the purpose of addressing

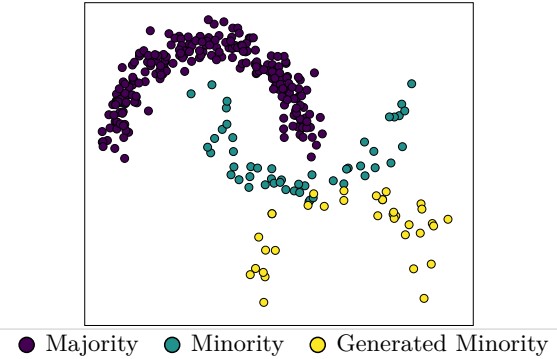

Figure 1: An M2m-style approach does not generate desirable samples on low-dimensional synthetic datasets, such as the two-moons toy dataset provided by scikit-learn (Pedregosa et al., 2011).

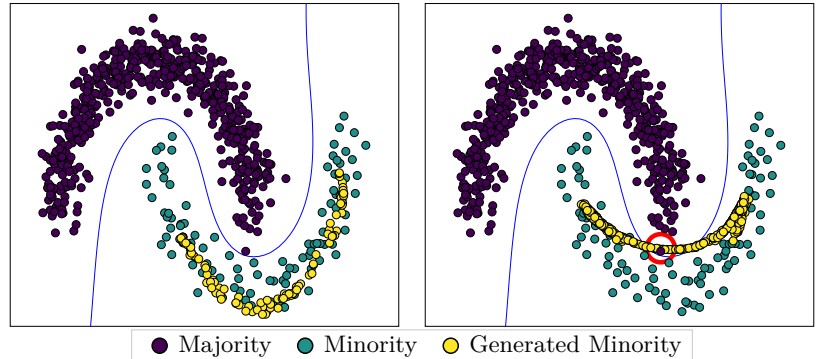

Figure 2: Samples are drawn from a two-moons toy dataset based on scikit-learn (Pedregosa et al., 2011). On the left, the new samples are generated by a standard ("vanilla") GAN which is trained to model the minority class distribution. On the right, the tabular translation GAN with $\lambda_T = 0.1, \lambda_C = 0, \lambda_I = 0$ demonstrates the ability of the translation loss to guide the model to generate samples that are close to the class boundary: the mean distance from the decision boundary (the blue curve) to the generated points is 0.3305 for vanilla GAN and 0.2173 for TTGAN. In the region highlighted by the red circle, some of the generated samples fall within the boundary of the majority class.

class imbalance: Kim & Byun (2020) apply it on petrophysical facies, Bar-El et al. (2021) classify X-ray images, and Zhu et al. (2018) perform emotion classification. Translation-based approaches to oversampling on tabular data are relatively underdeveloped. The most comparable approach to our scheme is M2m (Kim et al., 2020), where samples are translated by traversing along the path of the gradient of an initial baseline classifier. As the authors demonstrate, adding these translated samples improves classification performance on the tested datasets. In practice, the authors find that many of the synthetic points are adversarial samples: they are close to the input sample in the high-dimensional feature space. During our experimentation, this approach has led to unsatisfactory results in generating informative minority samples on *tabular* datasets, including toy synthetic datasets. We hypothesize that this is due to the feature heterogeneity and low dimensionality of tabular datasets. We then attempted to modify the approach by making more flexible gradient steps: instead of using a constant stepsize as used in the original paper, we adjusted the gradient stepsize by using optimization methods (Kingma & Ba, 2014). This did not yield significantly better results (see Figure 1 for an example of samples generated by M2m).

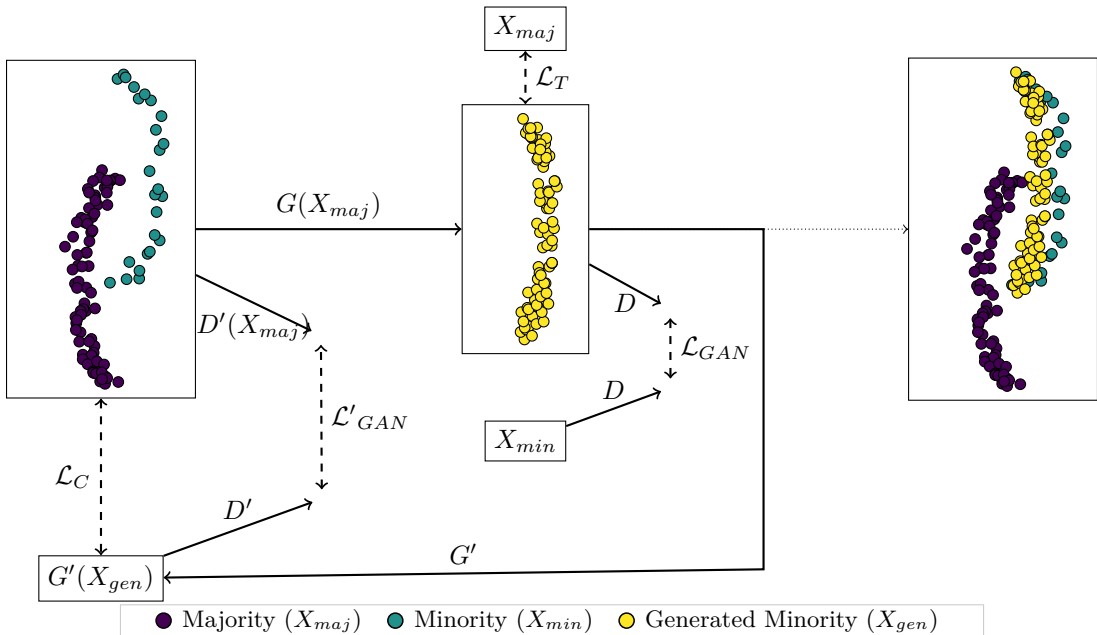

Figure 3: Training the tabular translation GAN involves imposing additional objectives: translation loss $\mathcal{L}_T$ (Eq. 3) encouraging closeness between input and synthetic samples, cyclic loss $\mathcal{L}_C$ (Eq. 4) encouraging invertibility for the generator, and identity loss $\mathcal{L}_I$ (Eq. 5) encouraging stability on the output distribution. These losses help constrain the position of the generated samples with respect to the majority samples. The generated samples are then included in the oversampled dataset. The identity loss $\mathcal{L}_I$ is omitted for the sake of clarity of the diagram.

## 2 Tabular Translation GAN (TTGAN)

### 2.1 Formulation

Given a binary imbalanced classification dataset

$$\mathcal{D} = \{(x_i, y_i)\}_{i=1}^N \text{ where } x_i \in \mathbb{R}^d \text{ and } y_i \in \{0, 1\},$$

we wish to generate new samples to include in an augmented dataset $\mathcal{D}'$. We interchangeably write $\mathcal{D} = X_{maj} \cup X_{min}$ where $X_{maj} = \{x_i | y_i = 0\}$ and $X_{min} = \{x_i | y_i = 1\}$ are the majority class and minority class respectively.

We refer to the set of synthetic samples we generate as $X_{gen}$ and select a subset $X_{selected}$ of those to add, such that $\mathcal{D}'$ is given by

$$\mathcal{D}' = \mathcal{D} \cup X_{selected} \text{ for } X_{selected} \subseteq X_{gen}.$$

We then fit a classifier $f$ (which may in principle be of any type) on $\mathcal{D}'$ and use it to classify test samples.

Table 1: Notable differences between the proposed TTGAN, vanilla GAN, and CycleGAN

|  | Vanilla GAN | CycleGAN | TTGAN |
|---|---|---|---|
| Networks | $G, D$ | $G, D, G', D'$ | $G, D, G', D'$ |
| Input training dist. | $z \sim Z$ | $z \sim X_{maj}$ | $z \sim X_{maj}$ |
| Generator loss func. | $\mathcal{L}_G$ (Eq. 2) | $\mathcal{L}_G + \lambda_C \mathcal{L}_C(z) + \lambda_I \mathcal{L}_I(z)$ | $\mathcal{L}_G + \lambda_T \mathcal{L}_T(z) + \lambda_C \mathcal{L}_C(z) + \lambda_I \mathcal{L}_I(z)$ |
| Synthesized samples | $X_{gen} = G(Z)$ | $X_{gen} = G(X_{maj})$ | $X_{gen} = G(X_{maj})$ |

## 2.2 Generative Adversarial Networks (GAN)

Goodfellow et al. (2014) introduced a breakthrough generative model to sample from an arbitrary distribution. GANs learn to generate new samples from a desired distribution using two neural networks, the generator $G$ and discriminator $D$, that play a min-max game where the generator is tasked with generating synthetic samples that fool the discriminator, which in turn is tasked with distinguishing between real and generated samples.

This minimax loss can be formulated as

$$\mathcal{L}_{GAN} = E_x[\log\left(D(x)\right)] + E_z[\log\left(1 - D(G(z)\right)], \tag{1}$$

where $E_x$ is the expected value over real samples $x$, and $E_z$ is the expected value over input noise to the generator $z$ which is sampled from a latent noise space $z \sim Z$ where $Z$ is some prior parametric distribution.

We note that the portion of the loss that is relevant to the generator is

$$\mathcal{L}_G = E_z[\log\left(1 - D(G(z))\right)], \tag{2}$$

which the generator tries to minimize and the discriminator tries to maximize.

We denote the portion that is not directly affected by the generator as

$$\mathcal{L}_D = E_x[\log\left(D(x)\right)],$$

which the discriminator tries to maximize.

While effective for sampling from the distribution, the standard GAN model provides no mechanism for controlling the location of a generated output sample given a particular input sample. In particular, this means that it is not designed to generate samples for the purpose of improving downstream imbalanced classification performance.

The introduction of the regularizing loss functions described below allows us to achieve this goal by performing translation from the majority class. The first such loss directly links between the output sample and the input sample. The additional losses, motivated by CycleGAN (Zhu et al., 2017), link between the output sample and the input sample by way of an additional GAN mapping from the minority to the majority.

## 2.3 Direct translation loss

For any vector $z$, we define the translation loss as

$$\mathcal{L}_T(z) = \|z - G(z)\|_1. \tag{3}$$

In order to guide the GAN to effectively translate input points to a close synthetic minority sample, we propose the following modifications to the base GAN architecture:

- Feeding real majority class samples as inputs to the generator during the training process: $z \sim X_{maj}$ rather than $z \sim Z$. The input $z$ to the generator is no longer a latent random noise vector, but rather an input feature-space sample we wish to translate.

- Adding the direct $L_1$ regularization term $\lambda_T \mathcal{L}_T(z)$ (Eq. 3) to the generator's loss objective.

We use the majority samples as input to the generator to help it learn from the statistics and geometrical properties of the majority class to generate a more realistic synthetic minority sample. Since the generator is forced to transform the majority samples into minority samples, it is required to learn a transformation that relies on the characteristics of both classes. Furthermore, using real samples as input to the generator can help improve the stability of the GAN training process since the input vectors of the generator are deterministic.

Intuitively, the addition of the translation loss guides the model to balance its primary objective of modeling the minority distribution with the objective of minimizing the distance between the output translated sample and the input sample. The magnitude of the hyperparameter coefficient $\lambda_T$ modulates this balance. This mechanism encourages the output samples to be located close to the class boundary.

Samples close to the boundary have an important contribution to classification performance, as they are more likely to be misclassified than ones far from the boundary; samples that are far from the boundary frequently have little effect on classification (Han et al., 2005).

The proposed modification allows us to achieve the desired effect of translating majority samples to minority samples that are close to the decision boundary, as shown in the synthetic example in Figure 2 where this direct translation loss is the only active additional loss. This mechanism achieves this effect without training an additional mirror GAN, and so may be employed by itself when constrained with resources and dataset size. However, to further leverage the majority samples (beyond using them as inputs for the generator training), we complement the direct translation loss with auxiliary losses adopted from the CycleGAN (Zhu et al., 2017) model. This also ensures invertibility of the generator function, which aids in alleviating mode collapse.

## 2.4 Tabular translation GAN

In addition to the direct translation loss, we use loss functions adopted from the CycleGAN (Zhu et al., 2017) model. Typically used for translating between unpaired domains of images, CycleGAN introduces an additional pair of networks $G', D'$ responsible for sampling from the majority class distribution. The following losses are used:

**Cycle-consistency loss -** Encourages the two generators to be roughly inverse mappings of each other:

$$\mathcal{L}_C(z) = \|G(G'(x_{min})) - x_{min}\|_1 + \|G'(G(x_{maj})) - x_{maj}\|_1. \tag{4}$$

(where $z$ takes on values of $x_{min}$ or $x_{maj}$).

**Identity loss -** Encourages the generators to be roughly the identity mapping on inputs that are sampled from the output distribution:

$$\mathcal{L}_I(z) = \|G(x_{min}) - x_{min}\|_1 + \|G'(x_{maj}) - x_{maj}\|_1. \tag{5}$$

Intuitively, these losses encourage the translation mechanism to possess the following desirable properties: that by translating back and forth we should arrive (close to) back where we started, and secondly that no translation is needed when already in the desired output domain.

The full tabular translation GAN thus consists of generators $G, G'$ and discriminators $D, D'$ which are all fully-connected neural nets. The aforementioned losses are added to the standard GAN loss objectives, such that the full loss function is given by

$$\mathcal{L}'_G = E_z[\log\left(1 - D(G(z)) + \lambda_T \mathcal{L}_T(z) + \lambda_C \mathcal{L}_C(z) + \lambda_I \mathcal{L}_I(z)],$$

and the complete GAN loss is

$$\mathcal{L}_{TTGAN} = \mathcal{L}_D + \mathcal{L}'_G. \tag{6}$$

The model modifications thus far are described in Table 1. Figure 3 shows the architectural scheme of the training process.

## 2.5 Implementation details

In our experiments, the generator is a 4 layer-deep fully-connected network with hidden sizes of 64, 128, and 256 with SELU (Klambauer et al., 2017) nonlinearities. The discriminator is a 3 layer-deep fully-connected network with hidden sizes of 128 and 64 with SELU nonlinearities. Training follows the standard training

procedure for GANs, with iterated steps of real and synthetic points, and we use the Adam optimization method with a constant learning rate of $10^{-4}$.

### 2.5.1 Synthetic samples selection criterion

Having trained the tabular translation GAN, we are able to generate synthetic minority samples by applying the generator on the majority class: $X_{gen} = G(X_{maj})$.

Although in principle we could augment the dataset with the entirety of $X_{gen}$, under certain circumstances this set may contain samples that are too far from the class boundary (in either direction) and thus detrimental. In particular, the additional losses imposed by TTGAN encourage the model to generate samples closer to the class boundary, which have a higher chance to drift over into the majority (as shown in the synthetic dataset of Figure 2). We apply a tune-able selection criterion, described below, to determine which of the synthetic samples to filter out and which to retain.

We first fit a baseline classifier $f_b$ on the input dataset $\mathcal{D}$, such that $f_b$ is of the same model type and architecture as the desired final classifier. We use $f_b$ to score the likelihood of each of the samples of $X_{gen}$ belonging to the minority class, and we sort the elements of $X_{gen}$ according to this score in descending order. We then apply a cut-off limit $p_{max}$ to filter out samples which have too high of a score (implying they are far from the boundary and thus less useful). Then we further restrict the number of selected samples to be a desired multiple $s$ of the size of the minority class, specifically:

$$X_{selected} = sorted(\{x \in X_{gen} | f_b(x) \leq p_{max}\})[: s|X_{min}|].$$

We consider both the cut-off probability $p_{max}$ and the number of samples to retain $s$ (as a multiple of the size of the minority class) to be hyperparameters of the model (see Section 3 for details on tuning).

The resulting augmented, resampled dataset is then given by

$$\mathcal{D}' = X_{maj} \cup X_{min} \cup X_{selected}.$$

The complete proposed procedure is detailed in Algorithm 1.

---

**Algorithm 1** Classification with Tabular Translation GAN

---

**Input:** Binary classification dataset $\mathcal{D} = X_{maj} \cup X_{min}$
**Output:** Classifier $f$
 1: Fit baseline classifier $f_b$
 2: Train tabular translation GAN using $\mathcal{L}_{TTGAN}$ yielding generator $G$
 3: Translate $X_{gen} \leftarrow G(X_{maj})$
 4: Filter synthetic samples
    $sort(\{x \in X_{gen} | f_b(x) \leq p_{max}\})[: s|X_{min}|]$
    and assign the result to $X_{selected}$
 5: Augment (oversample) dataset $\mathcal{D}' \leftarrow \mathcal{D} \cup X_{selected}$
 6: Fit final classifier $f$ on augmented dataset $\mathcal{D}'$

---

## 3 Experiments

We evaluate the classification performance of the proposed model under several different regimes of classifier model type, dataset, and imbalance ratio.

The alternative approaches that we compare to are: Re-weighting (RW) - the baseline approach where the chosen classifier is trained with re-weighting applied on the loss function to maximize performance on a balanced testing criterion, Random Oversampling (ROS) where the minority class is randomly oversampled so as to match the size of the majority class, SMOTE (Chawla et al., 2002), Borderline SMOTE (B-SMOTE) (Han et al., 2005), polynom-fit-SMOTE (Gazzah & Amara, 2008) (P-SMOTE), SUGAR (Lindenbaum et al.,

2018) and CTGAN (Xu et al., 2019). Re-weighting is then applied across all methods if a class imbalance remains.

For each experiment, we record values of the mean average precision (mAP) metric. In the result tables, we also note characteristics of the datasets ($N$ = # of samples, IR = imbalance ratio). For the tabular translation GAN model, the choice of hyperparameters was determined through sampling-based optimization on a validation set by using Optuna (Akiba et al., 2019). The search space for the CycleGAN losses was derived from the CycleGAN paper, and a wide search interval was given for the other hyperparameters: the translation loss weight $\lambda_T$, cut-off probability $p_{max}$ and ratio of samples to retain $s$. The tuning specifications are detailed in Appendix A.

In all experiments, pre-processing consisted of normalizing the features and handling categorical features. In the first and second set of experiments, categorical features were one-hot encoded, while in the third set the categorical features were target-encoded using the Category Encoders (McGinnis et al., 2018) package. In addition, in the third set missing feature values were imputed with the mode of the feature, and the Yeo-Johnson transformation (Yeo & Johnson, 2000) was applied to features that were determined to follow a power law distribution - specifically, if over 90% of the data falls in less than 20% of the value range of the feature.

Table 2: Classification performance on small (KEEL) datasets

| Dataset characteristics | | | mAP (rank) | | | | | | | |
|---|---|---|---|---|---|---|---|---|---|---|
| Dataset | N | IR | RW | ROS | SMOTE | B-SMOTE | P-SMOTE | SUGAR | CTGAN | TTGAN |
| abalone9-18 | 4174 | 129 | 0.8364 (2) | 0.7936 (6) | 0.8025 (5) | **0.8744** (1) | 0.8204 (4) | 0.7415 (7) | 0.6925 (8) | 0.8304 (3) |
| abalone19 | 731 | 16 | 0.0081 (8) | 0.0217 (2) | 0.0212 (4) | 0.0124 (6) | 0.0206 (5) | 0.0121 (7) | 0.0213 (3) | **0.0363** (1) |
| glass-0-1-6__vs__2 | 192 | 10 | 0.1958 (8) | **0.3923** (1) | 0.3250 (4) | 0.3583 (2) | 0.2603 (5) | 0.3310 (3) | 0.2494 (6) | 0.2386 (7) |
| glass2 | 214 | 11 | 0.1942 (2) | 0.1517 (6) | 0.1557 (4) | 0.1520 (5) | 0.1446 (7) | 0.1008 (8) | **0.2052** (1) | 0.1909 (3) |
| glass4 | 214 | 15 | 0.3417 (6) | 0.3000 (8) | 0.3595 (3) | **0.4444** (1) | 0.3417 (6) | 0.3417 (6) | 0.3444 (4) | 0.3833 (2) |
| page-blocks-1-3__vs__4 | 472 | 16 | 0.7958 (5) | 0.7579 (7) | 0.7579 (7) | 0.7579 (7) | 0.8259 (3) | 0.8076 (4) | 0.8357 (2) | **0.8972** (1) |
| yeast-0-5-6-7-9__vs__4 | 528 | 9 | 0.7284 (2) | 0.5421 (7) | 0.6245 (6) | 0.6911 (4) | 0.7203 (3) | 0.5380 (8) | 0.6445 (5) | **0.7502** (1) |
| yeast-1__vs__7 | 459 | 30 | 0.1980 (8) | **0.4043** (1) | 0.3076 (5) | 0.3709 (3) | 0.3786 (2) | 0.2800 (7) | 0.3259 (4) | 0.2926 (6) |
| yeast-1-2-8-9__vs__7 | 947 | 22 | 0.2355 (7) | 0.2499 (6) | 0.2647 (3) | 0.2677 (2) | 0.2590 (4) | 0.2206 (8) | **0.2776** (1) | 0.2546 (5) |
| yeast-1-4-5-8__vs__7 | 693 | 14 | 0.0591 (8) | 0.1084 (5) | 0.1589 (4) | **0.2416** (1) | 0.1821 (2) | 0.0793 (7) | 0.1064 (6) | 0.1294 (3) |
| yeast-2__vs__4 | 514 | 9 | **0.8563** (1) | 0.8467 (2) | 0.7892 (5) | 0.7540 (7) | 0.8067 (4) | 0.7080 (8) | 0.7827 (6) | 0.8162 (3) |
| yeast-2__vs__8 | 482 | 23 | 0.5294 (8) | 0.5481 (5) | 0.5450 (6) | 0.5482 (4) | 0.5409 (7) | 0.5518 (3) | 0.5544 (2) | **0.5573** (1) |
| yeast4 | 1484 | 28 | 0.4550 (6) | 0.4525 (7) | 0.4310 (8) | 0.5738 (3) | **0.6044** (1) | 0.4989 (4) | 0.4558 (5) | 0.5770 (2) |
| yeast5 | 1484 | 32 | 0.7326 (2) | 0.6620 (8) | 0.6865 (7) | 0.6888 (6) | 0.7210 (3) | **0.7419** (1) | 0.6993 (5) | 0.7170 (4) |
| yeast6 | 1484 | 41 | 0.5986 (3) | 0.4959 (6) | 0.5095 (5) | 0.3818 (8) | 0.5418 (4) | 0.4504 (7) | 0.6023 (2) | **0.6278** (1) |
| Mean | | | 0.451 | 0.448 | 0.449 | 0.474 | 0.478 | 0.427 | 0.453 | **0.487** |
| Mean rank | | | 5.067 | 5.133 | 5 | 4 | 4 | 5.867 | 4 | **2.867** |
| Median rank | | | 6 | 6 | 5 | 4 | 4 | 7 | 4 | **3** |
| # of datasets best | | | 1 | 2 | 0 | 3 | 1 | 1 | 2 | **5** |
| # of datasets worst | | | 5 | 2 | 1 | 1 | 0 | 4 | 1 | **0** |

For the first set of experiments, we test the performance of a linear SVM classifier on datasets from the KEEL (Alcalá-Fdez et al., 2011) collection. The subset of datasets consists of those with imbalance ratio greater than 9, and datasets for which attaining near-perfect accuracy is trivial were discarded. The data splits provided were used. The scikit-learn (Pedregosa et al., 2011) implementation for linear SVM is used and loss weights are balanced accordingly to compensate for imbalance. Examining the results in Table 2, we find that the proposed model outperforms all others in terms of mAP and median rank, performs the best on the highest number of datasets and is not the worst-performing approach on any dataset.

Table 3: Classification performance on a large (CelebA) dataset

| Dataset characteristics | | | mAP | | | | | | | |
|---|---|---|---|---|---|---|---|---|---|---|
| Dataset | N | IR | RW | ROS | SMOTE | B-SMOTE | P-SMOTE | SUGAR | CTGAN | TTGAN |
| CelebA | 203k | 40 | 0.4431 | 0.3918 | 0.3885 | 0.3773 | 0.4030 | 0.4062 | 0.39 | **0.4444** |
| CelebA (extreme imba.) | 199k | 200 | 0.1435 | 0.1243 | 0.0944 | 0.0981 | 0.1289 | 0.1254 | 0.1094 | **0.1445** |

The second set of experiments involves applying the state-of-the-art Catboost (Prokhorenkova et al., 2018) boosted trees model on a version of the large CelebA dataset where the features are annotated attributes rather than pixel values, and the target is another attribute (Liu et al., 2015). We also use an artificially extreme-imbalanced version of this dataset derived by withholding minority samples. Catboost is trained with the F1 loss function, a learning rate of 0.2, for 50 iterations with depth 3 trees. The results are recorded in Table 3. Here we see a clear advantage of the robustness of the proposed approach when compared to the alternatives: while the other approaches suffer degradation compared to the baseline approach, the tabular translation GAN instead provides a marginal performance improvement over the strong baseline model.

Table 4: Classification performance on huge (Playtika) datasets

| Dataset characteristics | | | mAP | | AUC-ROC | | Precision @ 0.4 Recall | |
|---|---|---|---|---|---|---|---|---|
| Dataset | N | IR | Baseline | TTGAN | Baseline | TTGAN | Baseline | TTGAN |
| Task 1 | 6.3M | 23 | 0.255 | **0.266** | 0.852 | **0.856** | 0.276 | **0.282** |
| Task 2 | 45.3M | 200 | 0.12 | 0.12 | 0.928 | 0.928 | 0.132 | 0.132 |
| Task 3 | 45.3M | 200 | 0.215 | **0.219** | 0.946 | 0.946 | 0.239 | 0.239 |
| Task 4 | 60M | 220 | **0.093** | 0.092 | 0.918 | 0.918 | 0.104 | **0.105** |
| Task 5 | 60M | 220 | 0.127 | **0.132** | 0.946 | **0.947** | 0.105 | **0.113** |

In the third set of experiments, we tested our method on datasets collected by Playtika [1]. Playtika collected and preprocessed datasets over the course of several months. The purpose of these datasets is to perform several classification tasks concerning the prediction of user behavior. Each of these datasets is very large, ranging from 6.3 million samples to 60 million samples, and each have imbalance ratios ranging from 23 to an extreme ratio of 220. Improving upon the Catboost-based baseline on these challenging tasks directly affects the ability of the organization to better understand its customers and to take appropriate steps to improve Playtika's capabilities. In addition to mAP, we measure performance on the metrics of AUC-ROC and Precision @ 0.4 recall, as these are business-relevant metrics used for internal evaluation. In the results shown in Table 4, we see a noticeable improvement on the primary dataset of focus, Task 1. The other datasets exhibit performance that either remains the same or more modestly improves; these datasets are large with varying characteristics and exhibit extreme imbalance. We also note that the mechanism of synthetic sample selection was slightly modified during these experiments: instead of $p_{max}$ representing an upper bound, samples were selected to be as close as possible to $p_{max}$ on either side of it; this improved performance and the interpretation of $p_{max}$ as a parameter that constrains the sample threshold remains unchanged.

Lastly, we also use the UMAP (McInnes et al., 2018) dimensionality reduction technique to visualize samples from the yeast4 (Alcalá-Fdez et al., 2011) dataset to better understand classification improvement. We compare an identical number of synthetic samples generated by a vanilla GAN and TTGAN respectively. This visualization, shown in Figure 4, demonstrates that TTGAN leads the synthesized points to be closer to the class boundary than with a vanilla GAN. The mAP achieved by the vanilla GAN is 0.5352, while the mAP with TTGAN is 0.5770.

## 3.1 Ablation analysis

To further analyze the performance of the proposed Algorithm 1, we conduct an ablation test by iteratively removing components of the model: comparing TTGAN to TTGAN without the CycleGAN losses (cyclic and identity losses), and comparing that to a "vanilla" GAN model. We conduct these tests on the page-blocks-1-3_vs_4 dataset from (Alcalá-Fdez et al., 2011). For all experiments besides the baseline, the number of samples generated was 5.5x the number of original minority, as determined by a hyperparameter search on a validation set. The results are shown in Table 5, and show that removing components of the model degrades performance.

The results also show that TTGAN without the CycleGAN losses improves upon the vanilla GAN. We note that Table 2 suggests that the optimal hyperparameter choices for some datasets did not involve the

---

[1]https://www.playtika.com/

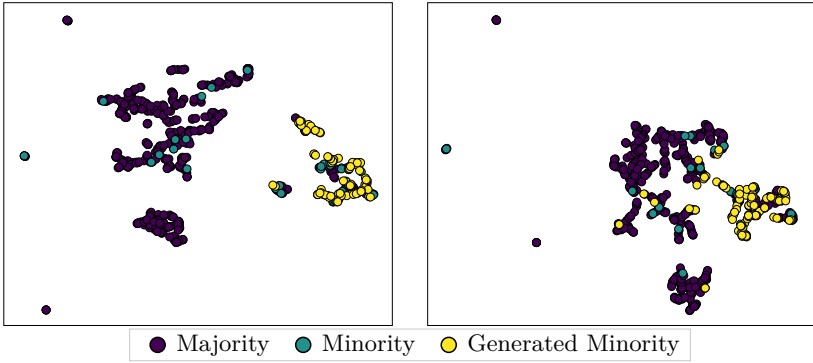

● Majority ● Minority ○ Generated Minority

Figure 4: A 2-dimensional visualization using UMAP (McInnes et al., 2018) of samples from the yeast4 dataset (Alcalá-Fdez et al., 2011). On the left, synthetic samples are generated by a vanilla GAN; on the right, synthetic samples are generated by TTGAN. TTGAN generates samples from more diverse regions, closer to the class boundary.

Table 5: Linear SVM page-blocks ablation analysis

| Model | mAP |
|---|---|
| Baseline | 0.7958 |
| Vanilla GAN | 0.8069 |
| TTGAN w/o cyclic+identity losses | 0.8357 |
| TTGAN | **0.8972** |

CycleGAN losses. We also note that all datasets in all experiments yielded hyperparameters such that the translation loss and cyclic loss were never simultaneously zero, despite that being an option in the hyperparameter search. This suggests improvement over a vanilla GAN trained on the minority class.

## 4  Conclusion

We presented an end-to-end method for performing imbalanced classification on tabular data, by introducing a GAN that is capable of translating majority samples to minority samples and then using it to generate useful synthetic samples of the minority class. The proposed GAN framework achieves this without requiring the use of a mirror GAN and cyclic loss - these components are optionally added to increase performance on certain datasets. We also described the mechanism by which we choose which of the translated samples we use. The experimental results presented show that this approach improves classification performance on a variety of datasets with different characteristics and with different types of underlying downstream classifiers, and that the results remains relatively robust without suffering from degradation.

In its current form, our model is designed to perform imbalanced binary classification on tabular datasets. The proposed approach could in principle be modified, applied and tested in other settings such as multi-class classification, or on different modalities of data. In addition, our model builds on GANs for generation, and so is vulnerable to any potential drawbacks of GANs, such as instability of training and mode collapse. Therefore, alternative generative models (such as Variational Auto Encoders (Kingma & Welling, 2013; Li et al., 2020), diffusion models (Ho et al., 2020) or normalizing flows (Kobyzev et al., 2020)) could perhaps be modified with our proposed additional losses in place of a GAN.

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

## A   Appendix

The hyperparameter search space for TTGAN (the other compared methods shared the range of $s$, the number of generated synthetic samples as a multiple of the minority sample set size) is given in Table 6. The full hyperparameter specification for the first set of experiments is described in Table 7, and the hyperparameters for the third set of experiments is described in Table 8. Experiments were run on an RTX 2080 GPU.

Table 6: Hyperparameter search space

| Epochs | $\lambda_T$ | $\lambda_C$ | $\lambda_I$ | $s$ | $p_{max}$ |
|---|---|---|---|---|---|
| 250-3000 | 0-0.3 | 0-20 | 0-15 | 0.2-16 | 0-1 |

Table 7: TTGAN KEEL hyperparameters - w / Linear SVM

| Dataset | Epochs | $\lambda_T$ | $\lambda_C$ | $\lambda_I$ | $s$ | $p_{max}$ |
|---|---|---|---|---|---|---|
| abalone9-18 | 1150 | 0.1 | 0 | 0 | 4 | 0.8 |
| abalone19 | 250 | 0.05 | 10 | 5 | 16 | 0.9 |
| glass-0-1-6__vs__2 | 2500 | 0.05 | 15 | 0 | 5.5 | 0.9 |
| glass2 | 2500 | 0.1 | 0 | 2.5 | 6.5 | 0.8 |
| glass4 | 900 | 0 | 15 | 7.5 | 5.5 | 0.8 |
| page-blocks-1-3__vs__4 | 900 | 0.05 | 5 | 5 | 5.5 | 0.7 |
| yeast-0-5-6-7-9__vs__4 | 900 | 0 | 5 | 2.5 | 6.5 | 0.8 |
| yeast-1__vs__7 | 2500 | 0.05 | 0 | 0 | 1.3 | 0.8 |
| yeast-1-2-8-9__vs__7 | 1150 | 0.05 | 10 | 5 | 4 | 1 |
| yeast-1-4-5-8__vs__7 | 900 | 0.1 | 15 | 0 | 1.65 | 0.7 |
| yeast-2__vs__4 | 500 | 0.1 | 15 | 0 | 1.8 | 0.6 |
| yeast-2__vs__8 | 1300 | 0.05 | 10 | 10 | 4 | 0.6 |
| yeast4 | 1000 | 0.05 | 10 | 0 | 4 | 1 |
| yeast5 | 1450 | 0.05 | 0 | 0 | 4 | 0.6 |
| yeast6 | 2500 | 0.15 | 0 | 5 | 7.5 | 0.6 |

Table 8: TTGAN Experiment 3 hyperparameters - w / Catboost

| Dataset | Epochs | $\lambda_T$ | $\lambda_C$ | $\lambda_I$ | $s$ | $p_{max}$ |
|---|---|---|---|---|---|---|
| Task 1 | 700 | 0.2 | 20 | 12 | 0.33 | 0 |
| Task 2 | 500 | 0 | 4 | 6 | 0.215 | 0 |
| Task 3 | 700 | 0.05 | 16 | 0 | 0.24 | 0.85 |
| Task 4 | 700 | 0.05 | 10 | 6 | 0.25 | 0.5 |
| Task 5 | 700 | 0.25 | 10 | 3 | 0.75 | 0.5 |

