# OpenReview forum: "Imbalanced Classification via a Tabular Translation GAN"
_TMLR — Rejected by TMLR_

### Review · Reviewer_Li8D · 2023-01-29

**Summary Of Contributions:**

This submission studies the issue of class imbalance in binary classification tasks using tabular data. The authors propose a GAN-based framework, called TTGAN, for generating synthetic samples for the minority class. Building on the CycleGAN architecture, TTGAN incorporates a new translation loss from the majority to the minority class and a selection criterion for selecting synthetic samples that are close to the decision boundary. The proposed method is shown to produce robust results through experiments on various datasets.

**Audience:**

Yes

**Claims And Evidence:**

Yes

**Requested Changes:**

Please see the "Weaknesses" section for a discussion of areas where the study could be improved. It would be beneficial to address all of the identified weaknesses in a future revision.

**Strengths And Weaknesses:**

### Strengths
1. Class-imbalanced classification is a prevalent and significant challenge in machine learning.
2. The TTGAN framework is based on a popular approach of oversampling and its motivation is well-justified.
3. The effectiveness of TTGAN is demonstrated through a variety of experimental results.

### Weaknesses
1. The translation loss used in TTGAN is similar to the identity loss in CycleGAN, what is the novelty of this loss and how does it contribute to the performance of the method?
2. The TTGAN framework involves many hyperparameters, could you provide more information on how to determine appropriate values for these parameters in a practical application?
3. The study does not include an ablation study on the effect of important hyperparameters such as the number of synthetic samples generated for each task.
4. The study does not include experiments on more complex problems such as multi-class classification, where the decision boundary is more complex. I am curious to know if TTGAN can generate useful synthetic samples in such scenarios.

---

> ### Author Response · Authors · 2023-02-27
> **Review reply**
>
> We’re very thankful for the review and appreciate these constructive comments. In the following, we address the concerns raised in the Weaknesses section in order.
>
> #### 1
> The translation allows us to construct translated synthetic samples with only a single GAN - the minority class GAN - without requiring a mirror GAN for the majority distribution.
>
> As demonstrated in Figure 2, this loss achieves the goal of generated samples close to the decision boundary. Table 5 shows that this loss alone improves upon the re-weighting baseline and the vanilla GAN (TTGAN w/o cyclic and identity loss). Finally, Table 7 demonstrates that the optimal hyperparameters (according to the validation set) for several of the datasets involved exclusively using this translation loss, without the CycleGAN losses.
>
> #### 2
>
> Table 6 describes the search ranges for each of the hyperparameters on all of the datasets. We used Optuna to sample the hyperparameter space and evaluated on a validation set. Given the applicability of this search space on all of the datasets tested, we expect it to perform well for other practical applications. The number of Optuna samples to run through may be tuned depending on resources/time available for training.
>
> #### 3
>
> Following this suggestion, we ran an ablation experiment for the page_blocks dataset:
>
> [Plot](https://i.imgur.com/pLgmCKu.png)
>
> The plot demonstrates that there exists a range either side of the optimum of the number of synthetic samples with which good performance is obtained, before degradation occurs.
>
> #### 4
>
> In real-world scenarios where multiple classes are present, data is often long-tailed such that there are a few dominant classes that may collectively be designated semantically as the “majority” and small classes that we might collectively designate as the minority; for instance, Wheelus et al. 2018 (full citation in bibliography) work on cybersecurity explores a dataset where normal (majority) traffic consists of several different protocol classes, and attacks (minority) consist of different attack family classes. Applying our model in this instance would generate synthetic samples within the minority family of classes by translating from the majority family of classes.
> Because the final downstream classifier that is trained on the augmented dataset is unconstrained and independent from our model, it can easily be taken to be a multi-class classifier.

---

### Review · Reviewer_Br7r · 2023-02-06

**Summary Of Contributions:**

The paper proposes TTGAN (Tabular Translation GAN), which aims to generate minority-class samples to address class-imbalanced classification tasks in tabular data. The paper adds a translation loss in addition to the original CycleGAN framework, and tests the proposed method in three datasets. TTGAN outperforms baselines models on average.


**Audience:**

Yes

**Claims And Evidence:**

No

**Requested Changes:**

Please see above in Weaknesses and Questions

**Strengths And Weaknesses:**

Strengthes:
- The paper propose a very simple addition to the original CycleGAN to generate minority-class samples.
- TTGAN is evaluated across multiple tabular datasets.

Weaknesses & Questions:
- In the last paragraph of 2.3, the paper states "The proposed modification allows us to achieve the desired effect of translating majority samples to minority samples that are close to the decision boundary". But I don't see how Eq.3 alone can guarantee that G will generate a minority class sample, without any cyclic loss or identity loss. It seems to me that Eq.3 will only guarantee that G will accept a sample from X_major and generate an output sample that is identical to the input sample.

- I'm not expert in addressing class imbalances, but the choice of baselines seems rather unchallenging. All baselines are based on classical statistics except one (maybe two if counting SUGAR), namely CTGAN, which was proposed in 2019. It seems that there must be more recent and well-known baselines. The paper mentioned Darabi & Elor (2021), which can be used to conditionally generate minority classes.

- CycleGAN is also missing from the baseline. The only difference between CycleGAN and TTGAN is the use of the translation loss. The authors must empirically show the effectiveness of their contribution (i.e., translation loss).

- I don't see the reason behind using different combinations of metrics/baselines across Table 2, 3, 4, and 5. Both AUPRC (i.e. mAP) and AUROC are important metrics in binary classification, so they should be used in all tables. Also I don't see why all the baselines disappeared in Table 4. I don't see why Table 2, 3, 4, and 5 cannot be just simply merged (i.e. test all models on all datasets in terms of AUPRC and AUROC, then present it in a single table)

- The paper fails to perform statistical tests to verify that TTGAN is indeed outperforming the baselines. I suggest that all models are evaluated on all 23 datasets (15 from KEEL, 2 from CelebA, 5 from Playtika, 1 from page-blocks) in terms of both AUROC and AUPRC. Then there will be enough samples to perform T-tests to verify if the performance gaps between TTGAN and the baselines are statistically meaningful. However, it should be taken into account that using AUPRC for statistical tests is probably not the best idea, since each dataset has a different baseline AUPRC, whereas the baseline AUROC is always 0.5. Rank according to AUPRC, however, might be okay.

- Figure 4 is supposed to show that TTGAN generates minority samples that are close to the decision boundary. But using UMAP will compress the latent space in a non-linear way, which makes it hard for the viewer to infer the decision boundary. Why not use PCA?

---

> ### Author Response · Authors · 2023-02-27
> **Review reply**
>
> We’re very thankful for the review and appreciate these constructive comments. In the following, we address the concerns raised in the Weaknesses & Questions section in order.
>
> #### 1
> The GAN model is trained to model the minority distribution, such that the discriminator is tasked with distinguishing between minority class samples and the synthetic samples. Thus, the standard GAN loss enforces the synthetic samples to be within the bounds of the minority distribution.
>
> Eq. 3 is used in conjunction with this standard GAN loss; we apologize if this was unclear.
> #### 2
> Following this suggestion, we have performed experiments with the method proposed by Darabi & Elor (2021) on the KEEL dataset collection described in Table 2; we selected the VAE-Poly variant which the authors found to provide the strongest results, and searched through a range of different synthetic-majority proportion ratios on the validation set. The mean mAP was **0.455**, between that of P-SMOTE and CTGAN, and below TTGAN’s **0.487**. The detailed breakdown will be added to the table.
> #### 3
> An important distinction between the translation loss and the CycleGAN losses is that the translation loss allows us to construct translated synthetic samples with only a single GAN - the minority class GAN with translation - without requiring a mirror GAN for the majority distribution. Table 5 demonstrates that a model with only the translation loss active provides a benefit over the re-weighting and vanilla GAN baselines. Additionally, Table 7 demonstrates that the optimal hyperparameters (according to the validation set) for several of the datasets involved exclusively using this translation loss, without the CycleGAN losses.
>
> Additionally, we have added CycleGAN (TTGAN w/o translation loss) to the ablation experiment described in Table 5. CycleGAN achieved an mAP of **0.8774**, less than that of TTGAN with the additional translation loss.
>
> #### 4 & 5
> Work is currently in progress to re-run all experiments across datasets and baselines such that both AUPRC and AUROC is recorded and compared. Unfortunately due to the volume of datasets, models and training required, this is time-consuming and will be included in our final version.
>
> #### 6
>
> We have generated PCA plots for the page blocks dataset (in the yeast4 dataset for which UMAP was used the majority and minority classes are not sufficiently distinguishable visually in the first 2 principal components).
>
> [Vanilla GAN](https://i.imgur.com/vXcWLtG.png)
>
> [TTGAN](https://i.imgur.com/LgY6BbS.png)
>
> (Purple samples are majority, green are minority, and yellow are synthetic).
>
> These plots also demonstrate the capability of TTGAN to generate samples in more diverse regions that are closer to the decision boundary.

---

### Review · Reviewer_fd1R · 2023-02-08

**Summary Of Contributions:**

This paper tackles an imbalanced (binary) classification problem on the tabular dataset. The authors propose a tabular translation GAN, which translates the majority samples to synthesize the minority samples. Specifically, they propose 1) additional regularization loss into original cycleGAN loss and 2) selection criterion to retain the useful synthesized samples. Empirical results on various tabular datasets demonstrate the proposed method's effectiveness as it performs better than the existing baselines.

**Audience:**

No

**Claims And Evidence:**

No

**Requested Changes:**

Please see the comments in *Weaknesses* and resolve the corresponding concerns

**Strengths And Weaknesses:**

**Strengths**
- Overall, the writing is clear and easy to follow. The organization of the main draft is well-established.
- Imbalanced classification is an important and interesting problem to solve. Also, the tabular dataset is a popular input domain, but the class imbalance problem is less explored with this.

**Weaknesses**
- The authors present the translation loss (eq. 3) along with empirical evidence (Figure 3). They argue that it “allows us to achieve the desired effect of translating majority samples to minority samples close to the decision boundary” (lines 137-138). However, this might be the result of careful tuning of hyper-parameter \lambda_T, not finding the optimal solution of the proposed loss if \lambda_T goes to \infinite or zero, the synthetic samples are located in the majority region or minority region, respectively. Hence, the same effect can be achieved with the other simple baseline (e.g., re-weighting GAN losses and controlling its coefficient to minority class). The authors should provide more support for this component, e.g., a more intuitive explanation and insensitivity to \lambda_T to achieve the desired property (closeness to decision boundary)
- As denoted in Section 2.5.1, the authors only retain the synthetic samples with low confidence as a minority class to gather the samples near the decision boundary. However, this criterion could induce some failure cases; for example, insufficiently translated samples (hence, it is hard to say this is “translated”) or out-of-distribution samples (far from training distribution, hence the classifier failed to be generalized). The authors should provide intuitive and empirical support for how the proposed method does not suffer the above degenerated cases.
- Although the authors provide the ablation study in Table 5, the following essential ablations are currently omitted. 1) As presented in Tables 6-8, this method has many hyper-parameters that are carefully tuned for each dataset. As the optimal values are largely different between datasets, the authors should provide sensitivity to the choice of hyper-parameters to address the concern of the cost of tuning them. 2) Through the paper, the authors argue that the usefulness of synthetic samples is decided by their distance to the decision boundary. Although translation loss can implicitly achieve this goal (eq. 3), an essential component for this seems to be the selection criteria. Hence, the authors should include 1) Vanilla GAN with selection criteria and 2) TTGAN w/o selection criteria for more clear validation. Also, if the authors can compare the results with 3) other selection criteria, such as high confidence score, it will enhance the advantage of the proposed criteria.
- Although the agreement with the importance of binary classification problems, many real-world problems often require multi-class classifications. However, the proposed method is not applicable to such cases without a significant increase in computational costs. Is there any efficient way to extend the proposed method to multi-class classification problems?

---

> ### Author Response · Authors · 2023-02-27
> **Review reply**
>
> We’re very thankful for the review and appreciate these constructive comments. In the following, we address the concerns raised in the Weaknesses section in order.
>
> #### 1
> The GAN model is trained to model the minority distribution, and so we’re not sure how to interpret the proposed baseline “re-weighting GAN losses and controlling its coefficient to minority class”. We would appreciate clarification on this point.
> #### 2
> The GAN model is trained to model the minority distribution, such that the discriminator is tasked with distinguishing between real minority class samples and the synthetic samples. Thus, intuitively, the standard GAN loss enforces the synthetic samples to be within the bounds of the minority distribution. Practically, a minimum threshold is used such that synthetic samples that have a less than 0.02 probability (according to the baseline classifier) of being a minority class sample are discarded.
>
> Empirically, Figure 2 provides visual and quantitative evidence (mean distance from decision boundary) of the ability of the translation loss to correctly translate samples close to the decision boundary.
>
> Additionally, we have now computed Histogram Based Outlier Scores (Goldstein & Dengel, 2012) on the page blocks dataset, comparing the outlier score of the synthetic samples with respect to the minority class, and with respect to the majority class. The HBOS of the synthetic samples with respect to the minority class is **12.001**, and the HBOS of the synthetic samples with respect to the majority class is **18.704**.  A higher HBOS signifies a more anomalous histogram; thus, the synthetic samples are effectively translated from the majority to the minority distribution.
>
> #### 3
>
> We note that the experiments conducted with the ranges of s and p_max that we have chosen allow implicitly for a variety of selection criteria: when p_max=1 no samples are filtered out and s dictates the number of samples added. For a given constant s, lowering p_max allows for choosing samples with progressively lower minority-class confidence. Due to the vanilla GAN not having a loss that encourages closeness to the decision boundary, it has a tendency to generate samples that have high confidence of being minority samples (see for instance Figure 2), and no significant number of samples with probability <p_max of being minority to select from (unless p_max is very high).
>
> Additionally, we have performed an ablation experiment on the sensitivity toward the number of synthetic samples chosen (corresponding to different values for the multiplier s) on the page-blocks dataset:
>
> [Plot](https://i.imgur.com/pLgmCKu.png)
>
>
> The plot demonstrates that there exists a range either side of the optimum of the number of synthetic samples with which good performance is obtained, before degradation occurs.
>
> #### 4
> In real-world scenarios where multiple classes are present, data is often long-tailed such that there are a few dominant classes that may collectively be designated semantically as the “majority” and small classes that we might collectively designate as the minority; for instance, Wheelus et al. 2018 (full citation in bibliography) work on cybersecurity explores a dataset where normal (majority) traffic consists of several different protocol classes, and attacks (minority) consist of different attack family classes. Applying our model in this instance would generate synthetic samples within the minority family of classes by translating from the majority family of classes.
> Because the final downstream classifier that is trained on the augmented dataset is unconstrained and independent from our model, it can easily be taken to be a multi-class classifier.

---

### Decision · Action_Editors · 2023-03-29

**Recommendation:** Reject

**Comment:**

Overall, reviewers commend that the paper is tackling an important ML problem, the presentation is clear and the proposed TTGAN approach is well motivated. They also appreciate the variety of the experimental datasets.

However the two major concerns from reviewers are:
1. Novelty of TTGAN is not high (as CycleGAN plus regularisation). -- Note that novelty is not the main criterion for acceptance at TMLR so this is not a major concern for me.
2. Some essential ablation study missing and the sensitivity of the approach to some hyper-parameters. -- This is the major concern from my perspective.

Regarding point 2, the authors provided feedback including preliminary results in response to reviewers' questions, which is encouraging. But the reviewers felt unsure whether these additional results are enough to address their concerns, and how they would be included in the final version if the paper is accepted.

I personally think that TMLR is the right venue for this paper, but the reviewers' point on missing key ablations & comparisons needs to be addressed to warrant acceptance. Therefore I encourage the authors to incorporate reviewers' suggestions on experiments in revision and perhaps re-submit to TMLR or other suitable venues.

**Audience:**

Researchers working on tabular data and imbalanced data, as well as on generative models.

**Claims And Evidence:**

This paper proposes Tabular Translation GAN (TTGAN) to tackle the data imbalance problem in tabular data classification tasks. It builds on top of CycleGAN by adding an additional regularisation term in the loss function. The effectiveness of TTGAN is demonstrated through a variety of experiments.